# Bomb $^{137}$Cs in modern honey reveals a regional soil control on pollutant cycling by plants

J. M. Kaste 1✉, P. Volante 1 & A. J. Elmore 2

$^{137}$Cs is a long-lived (30-year radioactive half-life) fission product dispersed globally by mid-20th century atmospheric nuclear weapons testing. Here we show that vegetation thousands of kilometers from testing sites continues to cycle $^{137}$Cs because it mimics potassium, and consequently, bees magnify this radionuclide in honey. There were no atmospheric weapons tests in the eastern United States, but most honey here has detectable $^{137}$Cs at >0.03 Bq kg$^{-1}$, and in the southeastern U.S., activities can be >500 times higher. By measuring honey, we show regional patterns in the biogeochemical cycling of $^{137}$Cs and conclude that plants and animals receive disproportionally high exposure to ionizing radiation from $^{137}$Cs in low potassium soils. In several cases, the presence of $^{137}$Cs more than doubled the ionizing radiation from gamma and x-rays in the honey, indicating that despite its radioactive half-life, the environmental legacy of regional $^{137}$Cs pollution can persist for more than six decades.

---

[1] Geology Department, William & Mary, Williamsburg, VA, USA. [2] Appalachian Laboratory, University of Maryland Center for Environmental Science, Frostburg, MD, USA. ✉email: jmkaste@wm.edu

During the middle of the 20th century, five countries tested over 500 nuclear weapons in the air, which, taken together released far more ionizing radiation to the atmosphere than any other event or combination of events in human history[1,2]. The majority of these weapons were detonated in just a few locations in the northern hemisphere; the Marshall Islands in the Pacific Ocean (U.S.) and Novaya Zemlya (former U.S.S.R.) hosted over 75% of the energy yield of all the tests[3]. Many of the air detonations were so powerful that dozens of radioactive fission products (e.g., $^{137}$Cs, $^{90}$Sr, $^{131}$I) were injected into the stratosphere and distributed globally with a residence time of ca. 1 year before deposition primarily by rainfall. In 1963, the Nuclear Test Ban Treaty effectively limited atmospheric testing with the exception of a few dozen smaller yield detonations that occurred primarily in China and Africa in the 1960s and 1970s[1]. The presence of radioactive pollution from nuclear testing is globally ubiquitous, and detectable on every continent and even in deep ocean trenches[4]. Earth scientists widely apply the "bomb spike" radionuclides as a global chronological marker based on the assumption of strong radionuclide associations with minerals, and there are thousands of published studies using weapons fallout to trace soil transport and for dating sediments deposited in the 1960s[5,6]. The negative consequences of global nuclear fallout to human health are just recently coming into focus[7,8], but the long-term biogeochemical fate and ecological consequences of radioactive pollution from weapons tests in ecosystems outside the immediate vicinity of test sites is uncertain.

While most of the radiation produced by a nuclear weapon detonation decays within the first few days, one of the longest-lived and more abundant fission products is $^{137}$Cs, which has a radioactive half-life of 30.2 years. During radioactive decay to stable $^{137}$Ba, $^{137}$Cs atoms emit ionizing radiation, including beta particles with $E_{max} = 0.512$ MeV, Ba Kα x-rays at 0.032 MeV, and a 0.662 MeV gamma ray, and recent research suggests that even low amounts of $^{137}$Cs can kill organisms and disrupt essential ecosystem services[9-14]. Cesium is not required for plant growth and functioning[15]. However, because Cs has a similar ionic charge and radius as K, an essential element for plants, $^{137}$Cs is absorbed from the soil via K-specific membrane transporters in vegetation[16,17]. This pathway for $^{137}$Cs from soils to plants and into the human diet was predicted decades ago, leading to a government-supported radionuclide surveillance program that conducted widespread testing of milk in the late 1950s–1980s[18,19]. However, there is no published research documenting the presence or absence of $^{137}$Cs in eastern U.S. plants or in the U.S. food supply since 1988[20]. Here we present the first measurements of $^{137}$Cs in honey sourced from the eastern U.S. and leverage this with a high-resolution dataset of soil potassium which gives us the power to show regional patterns in the biogeochemical cycling of this dangerous radionuclide more than 50 years after deposition. We find that soil-plant-pollinator focusing effects can magnify $^{137}$Cs by several orders of magnitude in honey sourced from specific physiographic regions with low soil K.

Honey is produced by wild and managed pollinators around the world, and, because bees make this product by reducing the water content of flower-derived nectar by nearly 5-fold, environmental contaminants are naturally concentrated in this food[21-23]. In the years following the 1986 Chernobyl power plant failure, several researchers documented the presence of $^{137}$Cs in honey and pollen in European nations affected by fallout from the event[24-26]. The focus of these studies has been demonstrating that, with just a few exceptions, the $^{137}$Cs burden in honey was generally safe for human consumption[26], and, that honey is a useful indicator of atmospherically deposited contaminants and identifying modern pollution "hot spots"[23]. Moreover, previous studies of $^{137}$Cs in honey have focused on relatively small

geographic areas without much accompanying geospatial information of the pollinated region beyond suspected flower type or soil $^{137}$Cs burden, and virtually all of these have been in Europe where the source of $^{137}$Cs is a mixture of 1986 Chernobyl fallout and regional nuclear fallout from the 1950s to 1960s.

## Results and discussion

Eastern North America received disproportionally high fallout (4000–6500 Bq $^{137}$Cs m$^{-2}$) from the 1950s to 1960s nuclear weapons tests despite being relatively far from the detonation sites because of prevailing westerlies and high precipitation[27]. The $^{137}$Cs in this region's soil today is sourced nearly entirely (>90%) from weapons tests with little contribution from Chernobyl or Fukushima, and there are few published studies of $^{137}$Cs in honey in North America and none in the eastern U.S.[21]. Thus, $^{137}$Cs in honey from this region today is almost entirely due to the biogeochemical cycling of 55+ year-old legacy $^{137}$Cs by plants. Using low background gamma spectrometry, we found detectable $^{137}$Cs (≥0.03 Bq $^{137}$Cs kg$^{-1}$ = 10$^{5.94}$ atoms tablespoon$^{-1}$) in 68 of 122 distinct honey samples sourced from North America. Most of these samples came from private small-scale eastern U.S. honey producers where we identified the hive locations at the U.S. county scale (110 samples), for which average soil K concentration and $^{137}$Cs deposition rates are available[28,29] (Fig. 1). Atmospheric deposition models and direct soil measurements show that the northeastern United States received slightly higher burdens of $^{137}$Cs from the weapons tests than the southeastern United States[1,27]. Surprisingly, the geographic pattern of $^{137}$Cs in honey is not correlated with this regional gradient of fallout ($r^2 < 0.01$; $p > 0.5$). Of 40 honey samples collected from U.S. states north of Virginia, only 12 had detectable $^{137}$Cs, but 36 of 39 honey samples from Florida, Georgia, and South Carolina had detectable $^{137}$Cs (Fig. 1). For comparison to the east coast samples, we analyzed 5 honeys from larger commercial operations in the central U.S. where the hives were located primarily on

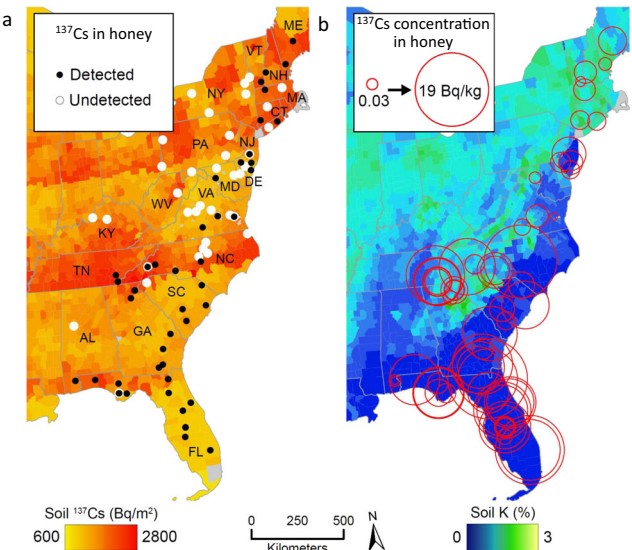

Fig. 1 Distribution of measured $^{137}$Cs in honey in the eastern United States for which county-scale information on the pollinator area was available ($n = 110$). a Detectable (filled black circles) $^{137}$Cs in honey on a map with 20th century $^{137}$Cs deposition to soils determined at the county scale[27,28] decay corrected to 2019. b Circles scaled logarithmically showing the relative magnitude of $^{137}$Cs in honey on a map in Bq/kg (becquerel=nuclear disintegration per second) with county mean soil K (potassium) concentrations determined from airborne radiometric surveys[29].

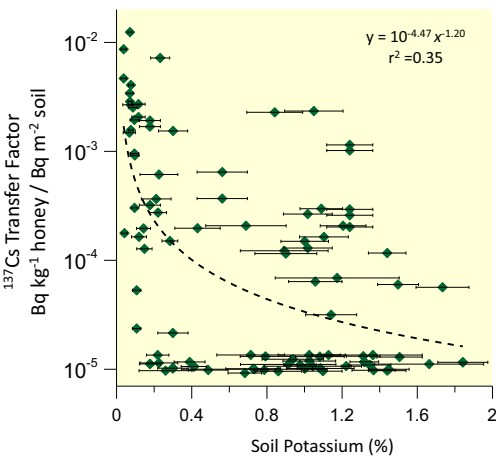

**Fig. 2 The transfer of $^{137}$Cs from soil to honey in the eastern United States scales inversely with the average soil K of the county from which it is sourced ($n$ = 106).** We tested for the effect of soil K on the $^{137}$Cs transfer factor on log normalized data and provide power function fit and $r^2$ adjusted for multiple comparisons ($p = 10^{-10.9}$). This analysis excludes the four honey samples from Florida specifically designated as agricultural. Uncertainty on this scaling is dominated by soil K variability within each county, shown here as one standard deviation of the mean. Honey with undetectable $^{137}$Cs are assigned half of the detection limit (0.015 Bq kg$^{-1}$) for the transfer factor calculation and statistical testing.

croplands. These honeys were very low in $^{137}$Cs, with 4 undetectable samples and one sample just at the detection limit, and a single sample from Cuba had undetectable $^{137}$Cs.

While soils of the eastern U.S. have a relatively narrow range of 1 to 2 kBq $^{137}$Cs m$^{-2}$ today, concentrations in honey sourced from this region spanned nearly 3 orders of magnitude with far higher levels in the southeast (Fig. 1b). We find an inverse relationship between the transfer of $^{137}$Cs from soils to honey and the mean total soil K concentrations of the county from it was sourced ($p < 0.001$; Fig. 2). While total soil K does not perfectly represent K availability to plants, a complete dataset of plant available K that would adequately describe the landscapes pollinated by bees producing the honey that we measured is not available. The largest dataset containing estimates of plant available K relying on ammonium acetate (NH$_4$Ac)-extractable K from soil samples is managed by the U.S. Department of Agriculture's National Soil Laboratory. However, these data are based on point measurements of soil chemistry and tend to be heavily focused in agricultural regions[30], while our honey was largely produced by bees pollinating undisturbed lands. The USDA database had NH$_4$Ac-extractable K data for just half of the counties that we had honey samples from, and these data also showed a similar pattern, with the logarithm of the $^{137}$Cs transfer from soil to honey scaling inversely with NH$_4$Ac extractable K ($p = 0.027$; $n = 55$; Supplemental Fig. 1). The airborne K data give far more continuous soil K coverage in the regions relevant to our study area[29], giving us the power to show patterns in biogeochemical cycling across this large and geologically diverse region (Fig. 2).

Our study reveals a regional pattern in the magnification of a legacy nuclear contaminant in honey ultimately controlled at the first order by geologic conditions and climate. The southeastern U.S. has relatively old, intensely weathered, and leached soils from the warm and wet climate on coastal plain geology, tending to be deficient in phosphorous, K, and other rock-derived nutrients[31]. In contrast, recent glaciation in the northeastern U.S. and freshly exposed bedrock in the Appalachian Highlands maintain a relatively large supply of K[32]. The climate and soil

parent material factors create a natural gradient of soil K that subsequently drives a regional control on $^{137}$Cs in honey, but agriculture causes local exceptions. Four of the honeys from FL that were specifically identified as sourced from managed orange groves or pepper crops, where K and N amendments are common averaged 0.2 Bq$^{137}$Cs kg$^{-1}$ compared with 18 FL wildflower honeys averaging 3.4 Bq kg$^{-1}$. This is consistent with our observations that agricultural honeys from the central U.S. tended to be low in $^{137}$Cs. Physiological differences across plant families also cause variations in Cs uptake[33], which likely impact our observations, but our hypothesis that soil K is the first-order regional control on the $^{137}$Cs content of honey is well supported by experimental-based research on Cs biogeochemistry[15,17,34–36].

**Soil potassium inhibits $^{137}$Cs uptake by vegetation.** Soil K depresses $^{137}$Cs uptake by plants and ultimately limits this radionuclide in vegetation via several mechanisms. Experiments using a range of plant species indicate that Cs is not required by vegetation and thus is never preferentially absorbed over K, rather, the Cs is taken because its ionic properties are near enough to K to allow mobility through K transport channels[15–17]. Increasing available K simply reduces Cs uptake due to the mass action relationships associated with plant-soil cation exchange reactions, thus K-based fertilizers are a proven method to reduce $^{137}$Cs uptake by food crops in heavily contaminated soils[37]. The ammonium ion (NH$_4^+$) also competes with Cs in these reactions, further explaining why honey sourced from managed agricultural systems tended to be very low in $^{137}$Cs. Kinetic controls on Cs uptake reactions may shift significantly at a threshold of K given that higher absorption rates by vegetation are observed in K limited soil[16]. Our measurements support a threshold model of $^{137}$Cs uptake into plants, given that all 26 honey samples collected from counties with soil K < 0.17% had detectable $^{137}$Cs, which included honey from 4 different U.S. States (Fig. 2). Based on the more limited dataset of NH$_4$Ac-extractable K[30] ($n = 55$), 13 out of 14 honeys from counties having <0.05 centimoles of exchangeable K$^+$ charge per kg had detectable $^{137}$Cs (Supplemental Fig. 1). Soil mineralogy likely plays an important role in sequestering $^{137}$Cs from plants. Soils high in K tend to have illite, a family of clay minerals with a strong capacity to absorb or include Cs, thereby reducing its bioavailability[38]. While the mechanisms limiting plant uptake of $^{137}$Cs have been identified theoretically and experimentally[34,35], our analyses of honey show how these processes play out regionally, controlling how plants and pollinators are exposed to order-of-magnitude differences in ionizing pollution based on local potassium concentrations.

The assumption that $^{137}$Cs is strongly fixed to soil and sediments forms the basis for its widespread use as sediment tracing and dating tool by earth scientists[6,39,40]. Thus, there is an absence of studies documenting the uptake of this fission product by native plants and possible pathways to the food supply in North America. One exception to this is the Pasteurized Milk Network (PMN), started in 1957 by the U.S. Public Health Service. Milk and honey are similar in that each is produced by foraging animals in every U.S. State, but in the 1950s, milk was considered the most likely pathway for nuclear fallout to enter the U.S. food supply. Consequently, a surveillance program sampled milk at central processing plants to monitor the radionuclide content, beginning with stations in five major cities[18], but expanding with individual state programs to include over 100 stations in the 1960s. With the PMN and individual state programs joining the efforts using the same methodology, over 10,000 measurements of $^{137}$Cs in milk are available, including monthly measurements from nearly every U.S. State for 1960 to 1975[19,41]. These data track the biological uptake of fission

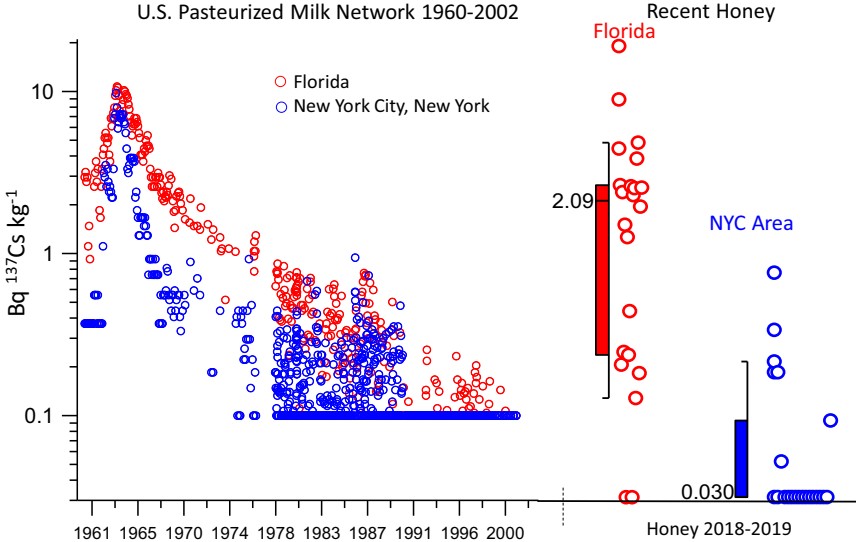

**Fig. 3 Monthly concentrations of $^{137}$Cs in milk measured by the U.S. Public Health Service Radionuclide Surveillance Program[19,41] from 1960 through 2002 in Tampa, Florida (red) and New York City (NYC, blue) compared with modern honey from Florida ($n = 22$) and the NY area ($n = 22$) on identical scaling.** The box center line indicates the median with the value given, the lower and upper bounds of the box show the 25th and 75th percentiles, respectively, and the minima and maximum whiskers give the 10th and 90th percentiles, respectively. These two regions have significantly different magnitudes of $^{137}$Cs contamination of honey (two-tailed Mann-Whitney test z score: 4.70, $p = 10^{-5.6}$). The detection limit is 0.1 and 0.03 Bq $^{137}$Cs kg$^{-1}$ for the milk and honey data, respectively. 20 out of 22 FL honeys had detectable $^{137}$Cs, but only 7 of the 22 NYC area (includes CT, NJ) honeys had detectable $^{137}$Cs.

products from soil to vegetation to milk in different regions of North America and are valuable to compare with our more recent measurements of $^{137}$Cs in honey.

Nationwide monthly average $^{137}$Cs concentrations in milk peaked in late 1963 at 6 Bq kg$^{-1}$ and fell sharply to <0.6 Bq kg$^{-1}$ by 1970 in response to the Nuclear Test-Ban Treaty[19]. Of the 122 honey samples we measured recently, three exceeded 6 Bq kg$^{-1}$, and 30 exceeded the 0.5 Bq kg$^{-1}$ concentration that nationwide average milk remained below after 1970. The highest honey $^{137}$Cs concentration that we measured from Florida in 2018 exceeded any reported monthly milk $^{137}$Cs value between 1958 and 2014, when the program formally ended (Fig. 3). This indicates that honey can be highly concentrated in $^{137}$Cs compared with other foods. The geographic pattern of $^{137}$Cs reported by the PMN is consistent with the pattern we find in honey today. Nearly every month between 1960 and 2014, the sampling stations in FL reported the highest $^{137}$Cs concentrations in milk compared with the rest of the United States. The $^{137}$Cs content of milk from FL had a delayed decline compared with NY area milk after 1963, and average honey $^{137}$Cs today in FL is significantly higher than average NY area honey (Fig. 3). Median soil K in FL is ca. 0.1%, compared with 1.3% in the NY area. Our data taken together with the PMN dataset show that plants growing in K deficient soils of the southeastern U.S. are more prone to absorbing $^{137}$Cs decades after atmospheric deposition, and that this contamination is transferred to foods by animal foragers.

To further investigate the widespread biogeochemical cycling of $^{137}$Cs from regional weapons pollution, we analyzed vegetation archives from the Hubbard Brook Experimental Forest (HBEF) and more recent collections to reconstruct $^{137}$Cs in native plants in the eastern U.S. over the last 50 years (Fig. 4). The HBEF one of the world's longest running ecosystem studies and maintains an archive of native foliage collected as early as the 1960s from several different northeastern states. These data show clearly that vegetation across multiple common species in the eastern U.S. has been declining in $^{137}$Cs. At its peak, however, the levels in vegetation during the 1960s through the 1980s were extremely

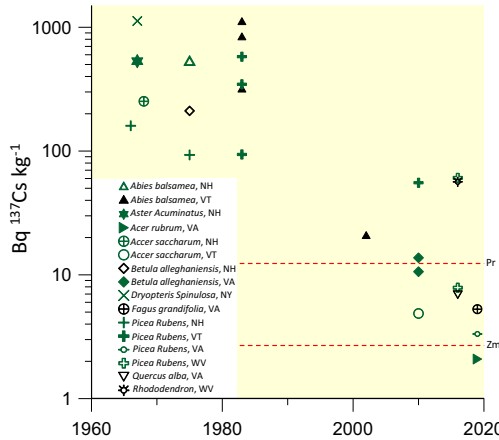

**Fig. 4 The $^{137}$Cs content of native vegetation (foliage) in the eastern U.S. using plant archives and more recent collections.** Each symbol represents a single plant, and the species and state are given in the legend. The dashed red lines Zm and Pr indicates the level of radiocesium in plant diets above which dose-dependent abnormalities and mortality have been observed in the pollinators *Zizeeria maha*[14] and *Pieris rapae*[48], respectively.

high, at 500 to over 1000 Bq kg$^{-1}$, which is orders of magnitude higher than levels that recent studies show may be dangerous for insects to digest[9,13,14]. $^{137}$Cs in vegetation declines from a median value of 390 Bq kg$^{-1}$ in the late 1960s to approximately 4 Bq kg$^{-1}$ in 2019, which is 2 orders of magnitude. The more high-resolution time series of $^{137}$Cs in milk (Fig. 3) shows similar scales of decline, from a 1963 high of 10.7 Bq kg$^{-1}$ in Tampa, F.L. and 9.8 Bq kg$^{-1}$ in N.Y. to values very near the 0.1 Bq kg$^{-1}$ PMN detection limit in both areas by 2010. These data taken together show how the biological uptake of $^{137}$Cs from soils declines significantly over time, as the radionuclide undergoes radioactive decay, but also as the $^{137}$Cs migrates down beneath the active plant rooting depth[5]. If the $^{137}$Cs content of honey followed a similar trend as milk (Fig. 3) and foliage (Fig. 4) then average

$^{137}$Cs in honey in the Southeastern U.S. (GA, FL, SC, NC) may have been far over 100 Bq kg$^{-1}$ in the 1960s–1970s, which even exceeds most national concentration standards for human consumption[42].

**A long and widespread environmental legacy from the bombs.** While the concentrations of $^{137}$Cs we report in honey today are below the 50–100 Bq kg$^{-1}$ dietary threshold level of concern observed by many countries[42], and not evidently dangerous for human consumption, the widespread residual radiation up to 19.1 Bq $^{137}$Cs kg$^{-1}$ is surprising given that nearly 2 half-lives have elapsed since most of the bomb production of $^{137}$Cs. All life on earth has naturally occurring sources of alpha, beta, and gamma radiation. In plants and thus most foods, this ionizing radiation will come primarily from $^{40}$K, which we measured concurrently with $^{137}$Cs in all honey samples. We calculate the ionizing radiation exposure rates to the honeybee from measured $^{137}$Cs and $^{40}$K concentrations in the honey using external dose coefficients published by the International Commission on Radiological Protection (ICRP, 2017)[43]. Given a typical 15 kg of honey in a hive[44], the total dose rate of ionizing radiation to bees from honey is <0.1 μGy hr-1, which is generally considered to be low[45] but the presence of $^{137}$Cs in honey is significant compared with background $^{40}$K in many cases (Fig. 5). In one case, the presence of $^{137}$Cs more than tripled the ionizing radiation in the honey, in 7 cases $^{137}$Cs contamination more than doubled it, and in 41 cases (33%) the $^{137}$Cs sourced >10% of the ionizing radiation in the sample. The 25 highest ranked $^{137}$Cs/$^{40}$K (Fig. 5) were from the southeastern US (FL, NC, TN, GA) where soil potassium is low.

In the last five years, it has become clear that insects suffer significant negative consequences at radiation dose rates that were previously considered safe, but the threshold at which damage occurs is debated[11–14,45–47]. Some studies indicate that low levels of $^{137}$Cs pollution can be lethal to pollinating insects and that any increase above background causes measurable damage to surrounding ecosystems[10,12–14,48]. Experiments with the Pale Blue Grass Butterfly, *Zizeeria maha* indicate a linear dose-dependent response in deformations and mortality when the insects eat vegetation ranging from 2.5 to 48 Bq Cs kg$^{-1}$ in Japan[13,14]. A significant fraction of honey samples collected from the southeastern United States (16 out of 49 south of VA) fell in

this range in 2018–2019, and decades ago it must have been far higher. Follow-up studies with a different species of butterfly, *Pieris rapae* also indicate that the ingestion of low-levels of radiocesium, even at levels less than naturally occurring $^{40}$K activity imposed biologically negative effects[48]. Populations of pollinating butterflies and bumblebees across a $^{137}$Cs pollution gradient near the Chernobyl Nuclear Power Plant in the Ukraine appear significantly reduced at even the lowest incremental exposure increases between 0.01 and 0.1 μGy h$^{-1}$. As the pollinator abundance declines around Chernobyl, significant impacts on ecosystem functioning were quantified through reductions in fruit production and tree recruitment[10]. If these studies represent the sensitivity of pollinating organisms to radiocesium pollution in the eastern United States, then $^{137}$Cs activities in native vegetation of 10 to >1000 Bq kg$^{-1}$ (Fig. 4) may have caused far more ecological damage than realized. Other recent studies, however, indicate that the negative effects of ionizing radiation to insects only begin above a threshold of 40 μGy h$^{-1}$, dose rates seen in the Chernobyl Exclusion Zone and below levels previously thought as harmless[45], but also far above the doses we quantify to bees from honey in the eastern U.S. today. Given that pollinating insects provide vital services to the world's ecosystem and are essential in maintaining global food security[49], more research is needed to help us better understand how ionizing pollution threatens their health and survival.

Several long-standing international agreements on nuclear nonproliferation and arms control have been dismantled in recent years, and some leaders have begun to put increased emphasis on the role of nuclear weapons in military strategies[50]. These developments may lead to renewed weapons testing activities by the nine nuclear capable nations and could encourage other nations to begin testing programs of their own. Moreover, given the efficiency and relatively low greenhouse gas footprint of nuclear energy for electricity generation[51], increased reliance on nuclear processes as a fuel source for domestic and military needs is likely in the coming years. Future releases of fission products to the environment are thus likely. We show that models can be developed to predict regional patterns where the uptake of $^{137}$Cs by vegetation will be enhanced and long-lasting, and our approach can guide new research on the biogeochemical fate of other common fission-products (e.g., $^{90}$Sr, $^{131}$I) in the environment.

## Methods

**Collections and analytical conditions.** 122 discrete honey samples were procured from beekeepers as raw, pure, and unfiltered. Honey samples were analyzed for radionuclides by directly photon-counting 50–200 grams of sample (depending on sample availability) using shielded high resolution (peak full-width half-maximum < 1.5 keV at 662 keV) intrinsic germanium gamma spectrometers typically for a minimum of two days but longer count times were often employed to reduce uncertainties. Gamma spectra were collected using Genie 2000 Software, version 3.2, Canberra Industries (2009). Detector efficiency for Canberra 5030 Broad Energy Detectors (150 cm$^3$ active volume) is determined by counting a certified liquid $^{137}$Cs source (Eckert & Ziegler) and high purity KCl (Fisher Scientific) for the different counting geometries. A typical detection limit determined by counting "blanks" for a 150 gram sample counted 200k seconds is 0.03 Bq kg$^{-1}$. We also scanned each honey spectrum for $^{134}$Cs, a strong gamma emitter at 605 keV and 796 keV, which given its short half-life of 2.06 years is a tracer of modern nuclear reactor leakage. Given that there was never a statistically significant photopeak at either 605 keV or 796 keV for any honey sample we conclude that the $^{137}$Cs we measure here is predominately legacy nuclear pollution associated with the weapons testing era. Concentrations of $^{137}$Cs and $^{40}$K along with two-sigma uncertainties based on counting statistics ($\sqrt{n}$) and uncertainty in the baseline subtraction are given in the Supplementary Data Table. The vegetation data shown in Fig. 4 are from foliage collected from the live plant either by our group directly (after 2010) or from vegetation archives maintained by the Hubbard Brook Experimental Forest.

**Soil potassium and Cs-137 data.** In 110 of the 122 samples, we identified the locations of the hives at the scale of the U.S. county or counties. Cs-137 fallout

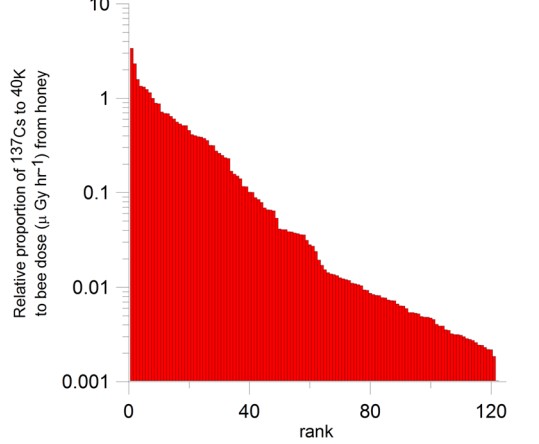

**Fig. 5 Ranked distribution showing the relative proportion of the ionizing radiation dose to honeybees from $^{137}$Cs contamination to background potassium radiation ($^{40}$K) using external dose coefficients[43] and measurements of these radionuclides in 122 honey samples.** Dose coefficients are used to convert measured radionuclide activities per mass to dose rates in units of micro (μ, 10$^{-6}$) Grays (Gy) per hour (hr).

depositional fluxes to soils have been calculated from 1953 to 1972 based on precipitation records and measurements of $^{90}$Sr in air and precipitation and match soil records reasonably well[1,27]. To determine county-averaged soil potassium, we use a high-resolution soil potassium, dataset determined by airborne radiometric surveys described in detail by the U.S. Geological Survey[29]. A separate dataset containing ammonium-acetate extractable K[30], widely considered to represent 'plant available K' was also tested but these data only covered half of the U.S. counties from which we had sourced honey (See Supplemental Data). For plotting data and statistical tests, "undetected" $^{137}$Cs samples were assigned a value of ½ the detection limit, = 0.015 Bq $^{137}$Cs kg$^{-1}$. Regression analyses were performed in Microsoft Excel, and we used PAST 4.0 for the Mann-Whitney tests (Paleontologia Electronica 4(10): 9pp).

**Reporting summary**. Further information on research design is available in the Nature Research Reporting Summary linked to this article.

## Data availability

All original data generated by this study are available as a supplemental dataset associated with the manuscript. All analytical results for $^{137}$Cs and $^{40}$K activities in the 122 honey samples are given along with 2-sigma analytical uncertainties in Table S1. Soil potassium concentrations and $^{137}$Cs deposition for each county is given, along with the standard deviation of the soil county potassium data.

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

## Acknowledgements

We thank the many "citizen scientists" who provided honey samples for this research. We are grateful to David Elmore, Stephen Norton, and Eric Davidson who provided thoughtful feedback on manuscript drafts, Emily Nastase who assisted with Fig. 1, and Steve Simon and the National Cancer Institute who provided the $^{137}$Cs fallout data. We thank Amey Bailey and the Hubbard Brook Experimental Forest for providing access to vegetation archives. This work was funded by a summer research grant from William & Mary's Provost Office and by a generous faculty award from Joseph Plumeri.

## Author contributions

Kaste supervised the project design, sample collection, and gamma spectrometry analyses, Volante acquired samples, prepared samples for analysis, and assisted with gamma spectrometry, Elmore executed the geospatial analysis. All three authors assisted with interpretations writing, editing.

## Competing interests

The authors declare no competing interests.
