## [Peer Review File · Nature Communications]

REVIEWER COMMENTS

Reviewer #1 (Remarks to the Author):

The authors used honey from the U.S. to monitor radiocesium in the environment. This approach is not new. In fact, the authors failed to acknowledge any of the more recent literature on this topic. Of the 39 hits (using SciFinder to search for "137Cs" and "honey"), I only checked the first page of results, but neither of the 7 more recent papers was cited. The use of honey as a bioindicator of pollution is state of the art and its applicability for the monitoring of environmental radionuclides has been demonstrated several times before. The correlation between soil potassium and cesium uptake by plants has been demonstrated numerous times previously (more than 100 publications, especially after nuclear accident research), and hence is not novel either.

The overall flavor of the paper is quite alarmist. The use of "bomb atoms" is not only very unconventional in the scientific community but is clearly intended to (negatively) resonate with the public. The statement "the environmental danger to honeybees from this fission product can persist for more than six decades" is not warranted by any means. Did the authors calculate the actual dose and compare them to reference values by the ICRP or IAEA? No. How could they then possibly come up with the conclusion "...declines in ecosystem functioning are predicted." Based on what? Gut feeling? It seems rather simplistic that the authors take the pale blue butterfly as a reference, concluding that their eating habits of a few Bq/kg alone is responsible for their increased mortality rate. The Fukushima evacuation zone represents a much higher air dose background, and even if radiation is responsible for their decline (which is questioned by everybody in the field except for the Mousseau and Moller groups), the intake of a few μBq is certainly not a risk factor for these insects.

For a proper dose discussion, it would be a good start to use the proper units. Sieverts are abbreviated Sv (and not S, which stands for siemens). Let alone that Sv are only used for humans, whereas for animals Gy has to be used.

Sentences such as "It has generally been assumed that outside the vicinity of the test sites, 137Cs fallout washed off vegetation and was immobilized by soil, and thus of no ecological concern." are utterly ridiculous. How about Japan after Fukushima? How about Chernobyl's fallout affecting activity levels in food in Northern and Central Europe until today? Of course, much of the radiocesium has been immobilized, but if the hypothesis of this study was the entity of 137Cs would be immobilized forever, the core concept of this study was put on feet of clay.

I acknowledge that the number of samples analyzed is fairly high and that the authors used state of the art radioanalytical techniques that resulted in fairly low limits of detection. However, due to the lack of novelty and because of exaggerated health/environmental claims, I cannot recommend this manuscript for publication in Nature Communications. This is rather for the Journal of Environmental Radioactivity, after the exaggerated claims ("dangerous") have been removed. The study does have value by showing the contamination pattern in honey within parts of the United States, but this is not enough for Nature Communications.

Reviewer #2 (Remarks to the Author):

This manuscript is important. It tracks the distribution of 137Cs across the Eastern US as a model for understanding the biogeochemical redistribution of this bomb-testing produced isotope. The primary analytical target is honey which is strongly concentrated nutrient-rich product. It serves as a marker for biological mobilization of bomb-produced radionuclides similar to cow milk which was monitored (with a particular focus on 90Sr) for decades after atmospheric testing to evaluate the potential for human consumption through the food chain. The authors find a good correlation between the location

of high concentrations of radionuclides in honey with that for milk.

The results show that there is measurable ^{137}Cs in honey depending on environmental circumstance, which sorts by location. One of the primary controls on mobilization has to do with the availability of K which Cs can mimic chemically. K is a required macronutrient for plants and animals. If it is in adequate concentration in soil K will suppress Cs uptake whereas if it is in low concentration plants will utilize Cs along with the K that it can scavenge. The authors find that indeed those areas with low K concentration are roughly correlated with higher Cs concentrations in honey, although there seems to be a threshold effect involved rather than a smooth relationship between soil K and Cs uptake/incorporation into honey.

The Eastern US can be broken into two regions based on its proximity to the area influenced by the last glaciation and those areas that did not get bulldozed by glaciers in the relatively recent past. In the northern region the soils were reset to ~ 15000 years ago whereas in the southern region many of the soils have much longer residence times and hence have been more strongly depleted in nutrients. From an a priori standpoint we expect that the added ^{137}Cs would be more competitive with K in the southern area. Indeed that seems to be the case. However the result is correlative and subject to a rather crude measure of soil K (more on that below), which makes a subsidiary observation particularly important. They find that within the same geographic area soils that are actively managed for agriculture and therefore likely to have elevated K via chemical (or organic) fertilization yield honey which has much lower (or undetectable) levels of ^{137}Cs . This observation seems key to the argument and could be more developed – in New England and New Jersey/Delaware there are opposing patterns of detect/nondetect in the same general area, are all these differences due to added fertilizer or are other factors in play.

The purpose of this manuscript is to raise an important point about the biocycling of dangerous radionuclides and their concentration in the environment. Even after two half lives ^{137}Cs has the potential to damage honey producing bees and other wildlife. Indeed as they note ^{137}Cs has been assumed to be tightly sorbed to surfaces within soil and not readily mobilized as an ion that could be utilized in the biosphere. The authors show well that it is mobilized and concentrated in ways that has the potential to damaging to the biosphere. It is one of a number of stressors that are creating havoc with the natural environment. This is reason enough for this manuscript to see timely publication.

Above I mentioned that the measure of soil K is relatively coarse – it relies on aerial surveys of K radioactivity. This tool measures all K regardless of its immediate availability to the biosphere (K can be tied up in feldspar and mica in ways that make it slowly available to the environment). As shown in Fig 1b and Fig 2 soil K concentrations are well differentiated between the two Eastern US regions described above. The error for K in Fig 2 is relatively large and does not directly include uncertainty regarding the availability of K to ecosystems. Fortunately the USDA NRCS National Soil Laboratory has a huge soil analysis data base that specifically measures available K (ammonium acetate extractable K) which could be accessed to verify that soil K concentration measured by airborne survey is correlated with K that is considered to be biologically available. Perhaps an argument can be made that the existing correlation is sufficient for the purpose of the ms, but the exploration into just what constitutes enough K to protect a local ecosystem from Cs mobilization would be interesting.

Oliver Chadwick
Santa Barbara
Oct 27, 2020

Reviewer #3 (Remarks to the Author):

The authors have presented a really elegant and important study. The findings of elevated Cs137 in

honey are important for understanding the cycling of these human-produced radionuclides, and for documenting potential ecological consequences for honey bees. While I am not a geochemist, and cannot really evaluate this in any great detail regarding the chemistry involved, the explanation the authors offer for the role of potassium in the cycling of Cs137 seems plausible and adequately supported by their data.

I believe that this manuscript will be of broad interest to many scientists, and that it is certainly timely. I would recommend publishing with a few very minor editorial changes as noted below.

Jim Pizzuto
Dept. of Earth Sciences
University of Delaware

Detailed comments keyed to the text:

1. Suggested revised first sentence of abstract: ^{137}Cs is a long-lived (30-year radioactive half-life) and dangerous fission product dispersed globally by mid-20th century atmospheric nuclear weapons testing. It has generally been assumed.....and thus ^{137}Cs is of no ecological concern.
2.and in the southeastern U.S., activities can be over 500 times higher.
3. It might be helpful to explain why the low K soils have high ^{137}Cs in the abstract.
4. Page 4, paragraph 2. Experiments using a range of plant species.....and thus is never...
5. The regression equation in Figure 2 is incorrect. The power function must have a negative exponent.
6. Materials and Methods: I would be interested in a brief explanation of how they can achieve uncertainties of ~ 0.01 Bg/kg for ^{137}Cs . This is about an order of magnitude better than we achieve in our lab using a "well detector". What are typical detection limits and minimum detectable activities?

Dr. James M. Kaste
Associate Professor of Geology
Environmental Science & Policy Program Director
William & Mary
Williamsburg, VA 23187

phone: (757) 221-2951
email: jmkaste@wm.edu

December 10th, 2020

Manuscript NCOMMS-20-36313-T “Bomb atoms in modern honey reveal a regional soil control on pollutant cycling by plants”. Below are detailed responses to the points raised by each of the reviewers, with reviewer comments in italics.

Reviewer 1 (R1)

***R1:** The authors used honey from the U.S. to monitor radiocesium in the environment. This approach is not new. In fact, the authors failed to acknowledge any of the more recent literature on this topic. Of the 39 hits (using SciFinder to search for “¹³⁷Cs” and “honey”), I only checked the first page of results, but neither of the 7 more recent papers was cited. The use of honey as a bioindicator of pollution is state of the art and its applicability for the monitoring of environmental radionuclides has been demonstrated several times before.*

We cited a 2017 review paper on contaminants in honey in our first sentence introducing honey on page 2 of our manuscript (Herrero-Latorre et al., *The use of honeybees and honey as environmental bioindicators for metals and radionuclides: a review*). We acknowledge that we could have given a deeper background on the use of honey as an environmental monitor, and now include a discussion of prior work with 4 additional (2 from 2020) studies of ¹³⁷Cs in honey in our revised manuscript (see lines 58-68). This new discussion of prior work allows us to clarify the novelty of our study in 3 ways: i) prior research on ¹³⁷Cs in honey is mostly in European countries where the ¹³⁷Cs is sourced from both Chernobyl and regional nuclear weapons testing fallout, thus the timescale of ¹³⁷Cs contamination in honey is unknown ii) prior studies demonstrate ¹³⁷Cs concentrations in honey without detailed spatial information; our novel work combining ¹³⁷Cs concentrations in honey and soil K gives us the unique power to show a regional pattern in how native vegetation cycles this legacy contaminant 50+ years after its introduction to the system, and iii) the eastern U.S. received disproportional high ¹³⁷Cs deposition from nuclear fallout because of prevailing winds and high precipitation, but there are no published values of Cs-137 in honey from this region. We are happy to make these points much clearer to the reader in the revised manuscript.

***R1:** The correlation between soil potassium and cesium uptake by plants has been demonstrated numerous times previously (more than 100 publications, especially after nuclear accident research), and hence is not novel either.*

We completely agree and pointed this out in our original manuscript on page 4 under the heading **Soil potassium inhibits ¹³⁷Cs uptake by vegetation** by immediately citing 4 studies which explain or demonstrate this theory. However, we advance this important concept in an important way: the vast majority of prior work demonstrating the suppression of ¹³⁷Cs by vegetation was done under small controlled plot experiments with potassium amendments (e.g., Shaw, 1993 cited in manuscript, Smolders et al. 1997 cited in revision) and/or ¹³⁷Cs applications. Our work shows how this mechanism of ¹³⁷Cs uptake suppression is controlled in the natural environment by regional geology and soil chemistry, becoming the first-order control on the ¹³⁷Cs contamination of honey across a vast area (2500 km length scale) 50 years after the pollution event. We clarify the prior research (adding 2 additional references) and the advance of our study more clearly in the revision (see lines 125-145).

Dr. James M. Kaste
Associate Professor of Geology
Environmental Science & Policy Program Director
William & Mary
Williamsburg, VA 23187

phone: (757) 221-2951
email: jmkaste@wm.edu

R1: *The overall flavor of the paper is quite alarmist. The use of “bomb atoms” is not only very unconventional in the scientific community but is clearly intended to (negatively) resonate with the public.*

No, our intention is not to “negatively resonate with the public”. The phrase “bomb carbon” is frequent in top scientific journals to identify modern ^{14}C pollution from the 1950s-1960s in soils and sediments. Our intention was to clearly identify the ^{137}Cs we measure in honey as ^{137}Cs from nuclear tests, **not** from reactor failures (e.g., Chernobyl, Fukushima). This distinction sets our study apart from most, in that our measurements trace biogeochemical processes with a timescale of 50+ years, compared with the majority of studies that look at plant uptake or honey in Europe a few years after Chernobyl, for example, where the ^{137}Cs is from a mixture of sources. To reduce any perceptions of alarmism, we change our title to “Bomb ^{137}Cs in modern honey reveals a regional control on pollutant cycling by plants” in the revision.

R1: *The statement “the environmental danger to honeybees from this fission product can persist for more than six decades” is not warranted by any means. Did the authors calculate the actual dose and compare them to reference values by the ICRP or IAEA? No. How could they then possibly come up with the conclusion “...declines in ecosystem functioning are predicted.” Based on what? Gut feeling? It seems rather simplistic that the authors take the pale blue butterfly as a reference, concluding that their eating habits of a few Bq/kg alone is responsible for their increased mortality rate. The Fukushima evacuation zone represents a much higher air dose background, and even if radiation is responsible for their decline (which is questioned by everybody in the field except for the Mousseau and Moller groups)*

This statement “the environmental danger can persist for more than six decades” is based on at least eight recent papers published in top scientific journals by multiple research groups separately studying Cs pollution in parts of Europe and Japan showing evidence that radioactive Cs is uniquely dangerous to pollinators. Following the Fukushima disaster in 2011, Hiyama et al. (Scientific Reports, 2012) showed severe abnormalities in the Pale Blue Grass Butterfly (*Zizeeria maha*) collected shortly after the event compared with pre-event collections. This research group followed up with another controlled study in which *Zizeeria maha* was fed diets of vegetation containing radioactive Cs at various levels, finding that deformations in female butterflies (Nohara et al., 2014) was linearly dependent on ingested radioactive Cs dose $>2.5 \text{ Bq/kg } ^{137}\text{Cs}$ concentrations in vegetation. An even more recent study using another butterfly species, *Pieris rapae* also concluded that “negative developmental and morphological effects were detected” following low dose radiocesium exposure (Taira et al., 2019 in *Scientific Reports*).

A significant fraction of the honey samples we analyzed from the southeastern U.S. (16 out of 49) had ^{137}Cs levels $\geq 2.5 \text{ Bq kg}^{-1}$, which is in the same range that negative effects have been observed (Nohara et al., 2014; Taira et al., 2019) for insect diets. Yes, today the doses are relatively low, but, given the fact that ^{137}Cs has a 30 year radioactive half-life, the levels would have been more than double in the 1970s-1990s. It’s important that we also present vegetation data from the eastern U.S. showing that native foliage here had ^{137}Cs activities on the order of 1000 Bq kg^{-1} in the 1960s - 1980s (Figure 4), which is high contamination by any standard and orders of magnitude more than the

Dr. James M. Kaste
Associate Professor of Geology
Environmental Science & Policy Program Director
William & Mary
Williamsburg, VA 23187

phone: (757) 221-2951
email: jmkaste@wm.edu

levels that have been shown can be a toxic diet for certain insects. Other research groups (Moller et al., 2013; Moller & Mousseau, 2009) also showed that low levels of ^{137}Cs exposure impacted pollinator services near the Chernobyl nuclear reactor. We acknowledge that there a debate in the literature about the effects of low-dose ^{137}Cs exposure on ecosystems, but Reviewer 1's statement "...is questioned by everybody in the field except for the Mousseau and Moller groups" is a falsehood in the purest sense.

R1: *For a proper dose discussion, it would be a good start to use the proper units. Sieverts are abbreviated Sv (and not S, which stands for siemens). Let alone that Sv are only used for humans, whereas for animals Gy has to be used.*

Thank you for pointing out the typo- we were referring to a study that presented their ground dose data in Sieverts (Sv). That has been fixed.

R1: *Sentences such as "It has generally been assumed that outside the vicinity of the test sites, ^{137}Cs fallout washed off vegetation and was immobilized by soil, and thus of no ecological concern." are utterly ridiculous. How about Japan after Fukushima? How about Chernobyl's fallout affecting activity levels in food in Northern and Central Europe until today? Of course, much of the radiocesium has been immobilized, but if the hypothesis of this study was the entity of ^{137}Cs would be immobilized forever, the core concept of this study was put on feet of clay.*

The hypothesis of our study certainly wasn't that ^{137}Cs would be immobilized forever. We make the specific statement, "this pathway for ^{137}Cs from soils to plants and into the human diet was predicted decades ago" early in the paper, describe past observations of ^{137}Cs in milk, and give relevant citations. However, it's very important to note that ^{137}Cs is widely used as a sediment tracing and dating tool (of the 1960s)- a technique which relies on the assumption that the ^{137}Cs is irreversibly adsorbed to soil and sediment grains. And 'widely used' cannot be overstated- there are literally thousands of published papers which apply ^{137}Cs to date lake sediments, peat bogs, coastal marshes, marine sediments, and as a soil tracer - all having the core assumption that ^{137}Cs is not mobile. Furthermore, please note that Reviewer 2 agreed with this very statement,

R2: *"Indeed as they note ^{137}Cs has been assumed to be tightly sorbed to surfaces within soil and not readily mobilized as an ion that could be utilized in the biosphere". So, there are contrasting views on the mobility of ^{137}Cs in various systems. Because we are short on space in the abstract, however, we deleted the statement that R1 took issue with.*

R1: *I acknowledge that the number of samples analyzed is fairly high and that the authors used state of the art radioanalytical techniques that resulted in fairly low limits of detection. However, due to the lack of novelty and because of exaggerated health/environmental claims, I cannot recommend this manuscript for publication in Nature Communications. This is rather for the Journal of Environmental Radioactivity, after the exaggerated claims ("dangerous") have been removed. The study does have value by showing the contamination pattern in honey within parts of the United States, but this is not enough for Nature Communications.*

Dr. James M. Kaste
Associate Professor of Geology
Environmental Science & Policy Program Director
William & Mary
Williamsburg, VA 23187

phone: (757) 221-2951
email: jmkaste@wm.edu

Indeed, our manuscript presents a large and carefully collected dataset on ^{137}Cs in honey for a large continental area in which published information is very sparse. We are the first to couple > 100 ^{137}Cs determinations with detailed geospatial information on soil potassium from where the honey was sourced from. Given that pollinating insects provide vital services to the world's ecosystem and are essential in maintaining global food security, every effort should be taken to safeguard their safety (Potts et al., *Nature*, 2016). We believe that our manuscript should be published in *Nature Communications* because we are the first to show a regional pattern in plant cycling of a dangerous radioactive contaminant 50+ years following its introduction to the soil-plant system. The data and interpretations we provide should be taken in the context of recent works showing that pollinators can be negatively impacted by low-dose radiocesium exposure.

Reviewer 2 (R2)

R2: *This manuscript is important. It tracks the distribution of ^{137}Cs across the Eastern US as a model for understanding the biogeochemical redistribution of this bomb-testing produced isotope. The primary analytical target is honey which is strongly concentrated nutrient-rich product. It serves as a marker for biological mobilization of bomb-produced radionuclides similar to cow milk which was monitored (with a particular focus on ^{90}Sr) for decades after atmospheric testing to evaluate the potential for human consumption through the food chain. The authors find a good correlation between the location of high concentrations of radionuclides in honey with that for milk.*

The results show that there is measurable ^{137}Cs in honey depending on environmental circumstance, which sorts by location. One of the primary controls on mobilization has to do with the availability of K which Cs can mimic chemically. K is a required macronutrient for plants and animals. If it is in adequate concentration in soil K will suppress Cs uptake whereas if it is in low concentration plants will utilize Cs along with the K that it can scavenge. The authors find that indeed those areas with low K concentration are roughly correlated with higher Cs concentrations in honey, although there seems to be a threshold effect involved rather than a smooth relationship between soil K and Cs uptake/incorporation into honey. The Eastern US can be broken into to two regions based on its proximity to the area influenced by the last glaciation and those areas that did not get bulldozed by glaciers in the relatively recent past. In the northern region the soils were reset to ~ 15000 years ago whereas in the southern region many of the soils have much longer residence times and hence have been more strongly depleted in nutrients. From an a priori standpoint we expect that the added ^{137}Cs would be more competitive with K in the southern area. Indeed that seems to be the case.

However the result is correlative and subject to a rather crude measure of soil K (more on that below), which makes a subsidiary observation particularly important. They find that within the same geographic area soils that are actively managed for agriculture and therefore likely to have elevated K via chemical (or organic) fertilization yield honey which has much lower (or undetectable) levels of ^{137}Cs . This observation seems key to the argument and could be more developed – in New England and New Jersey/Delaware there are opposing patterns of detect/nondetect in the same general area, are all these differences due to added fertilizer or are other factors in play

We are happy that Reviewer 2 clearly appreciates the importance of our study and acknowledge that there are opposing patterns of detected & non-detected ^{137}Cs in regions with similar K

Dr. James M. Kaste
Associate Professor of Geology
Environmental Science & Policy Program Director
William & Mary
Williamsburg, VA 23187

phone: (757) 221-2951
email: jmkaste@wm.edu

concentrations. There are a number of reasons that likely cause this: first, the potassium data we use does not perfectly represent plant-available potassium perfectly (as R2 points out himself). Second, is that we simply don't know precisely where the bees went to gather nectar. Because soils can be very chemical heterogeneous on 10 km scales with regards to clay content and nutrient status, it is simply possible that our county-scale (ca. 500 km scale) concentration didn't effectively capture the smaller region that the bees were pollinating. Finally, as we point out, clay mineralogy is a crucial control on Cs and K sequestration- "illite" type clays can irreversibly trap Cs and K in the interlayer; thus the mineralogy of the soils which we don't have data on at the appropriate scale can also play a role in regulating plant uptake of ^{137}Cs (Fuller et al., 2015). Thus, we show that soil K is a first-order regional control on the plant uptake of ^{137}Cs , but there are other interesting research questions to pursue further.

R2: *Above I mentioned that the measure of soil K is relatively coarse – it relies on aerial surveys of K radioactivity. This tool measures all K regardless of its immediate availability to the biosphere (K can be tied up in feldspar and mica in ways that make it slowly available to the environment). As shown in Fig 1b and Fig 2 soil K concentrations are well differentiated between the two Eastern US regions described above. The error for K in Fig 2 is relatively large and does not directly included uncertainty regarding the availability of K to ecosystems. Fortunately the USDA NRCS National Soil Laboratory has a huge soil analysis data base that specifically measures available K (ammonium acetate extractable K) which could be accessed to verify that soil K concentration measured by airborne survey is correlated with K that is considered to be biologically available. Perhaps an argument can be made that the existing correlation is sufficient for the purpose of the ms, but the exploration into just what constitutes enough K to protect a local ecosystem from Cs mobilization would be interesting.*

We agree- the potassium dataset we chose to use has the highest spatial resolution and coverage (Smith et al., 2013 in *Geoscience Frontiers*), but it doesn't perfectly represent plant available K. We carefully checked USDA NRCS dataset and found that there are NH_4Ac -extractable potassium data (considered by many as "plant available potassium") in 55 of the counties that we had honey from- that's about half of our dataset. The log of the transfer factor of ^{137}Cs from soil to honey had the same significant inverse trend with NH_4Ac -extractable K ($p=0.027$) as seen for total K (Fig 2) but the significance is higher when we use the total K data because $n=106$. We provide this new analysis of ^{137}Cs in honey with "plant extractable K" in Supplemental Fig. 1. There are "pros" and "cons" to each dataset and give a discussion on this in the revised manuscript (lines 100-114; 150-152).

Reviewer 3 (R3)

R3: *The authors have presented a really elegant and important study. The findings of elevated Cs137 in honey are important for understanding the cycling of these human-produced radionuclides, and for documenting potential ecological consequences for honey bees. While I am not a geochemist, and cannot really evaluate this in any great detail regarding the chemistry involved, the explanation the authors offer for the role of potassium in the cycling of Cs137 seems plausible and adequately supported by their data.*

I believe that this manuscript will be of broad interest to many scientists, and that it is certainly timely. I would recommend publishing with a few very minor editorial changes as noted below.

Dr. James M. Kaste
Associate Professor of Geology
Environmental Science & Policy Program Director
William & Mary
Williamsburg, VA 23187

phone: (757) 221-2951
email: jmkaste@wm.edu

We are happy to hear that reviewer 3 believes that the manuscript will have broad interest and is timely. We appreciate the Reviewer's ideas on editorial changes below:

R3: *Suggested revised first sentence of abstract: ^{137}Cs is a long-lived (30-year radioactive half-life) and dangerous fission product dispersed globally by mid-20th century atmospheric nuclear weapons testing. It has generally been assumed.....and thus ^{137}Cs is of no ecological concern.*

Edit made as suggested for the first fragment, the second fragment refers to a sentence that has been deleted.

2.and in the southeastern U.S., activities can be over 500 times higher.

Edit made as suggested.

R3: *It might be helpful to explain why the low K soils have high ^{137}Cs in the abstract.*

Given the 150 word limit, we don't have space to do this. However, we provide a clear explanation of this in the manuscript.

R3: *Page 4, paragraph 2. Experiments using a range of plant species.....and thus is never...*

Edit made as suggested

R3: *The regression equation in Figure 2 is incorrect. The power function must have a negative exponent.*

We appreciate that the Reviewer caught this typo. A revised figure with an accurate power function equation is given in the revision.

R3: *Materials and Methods: I would be interested in a brief explanation of how they can achieve uncertainties of ~ 0.01 Bg/kg for Cs^{137} . This is about an order of magnitude better than we achieve in our lab using a "well detector". What are typical detection limits and minimum detectable activities?*

Our uncertainties and detection limits are relatively low for several reasons. First, we use a large active volume planar high-purity intrinsic Ge detector (150 cm^3) which has a high stopping power for the ^{137}Cs gamma ray. The absolute detector efficiency for the 662 keV gamma-ray for a 105 mL geometry sample is 3.5%, and the system is designed with an ultra-low background cryostat and remote detector chamber to keep the background low. Furthermore, we analyze large sample masses (typically 150 grams) and employ long count times (2+ days per sample), which increase our ability to resolve the signal from noise at 662 keV. In practice, the detection limit varies from sample to sample because of differences in the background from Compton scattering by gamma rays emitted by other radionuclides, but our typical background uncertainty in the system ascertained by

Dr. James M. Kaste
Associate Professor of Geology
Environmental Science & Policy Program Director
William & Mary
Williamsburg, VA 23187

phone: (757) 221-2951
email: jmkaste@wm.edu

200 ksec+ background counts for a 150 gram sample is 0.03 Bq/kg. Average uncertainty in our 122 samples was 0.06 Bq/kg, with lower uncertainties for samples counted longer. The well-type detector referenced by R3 has high efficiency (up to 7% typically for ^{137}Cs) for small sample masses (<3 grams), but the lower analyzed mass causes uncertainties and detection limits to be far higher than the configuration we use here. We added details in the methods section to better explain this.

Again, we would like to thank the reviewers for their thoughtful evaluations and suggestions. We trust that you will agree that the revisions we made based on the reviewer comments have greatly strengthened the manuscript. Thank you for your consideration and please do not hesitate to contact me if you have any questions about our work.

Sincerely,

on behalf of co-authors Andrew Elmore and Paul Volante

REVIEWERS' COMMENTS

Reviewer #1 (Remarks to the Author):

The paper lacks the necessary discussion of doses to justify claims such as the suggestion that “environmental danger” persists for more than six decades or “the magnification of ^{137}Cs in honey should be carefully considered as a major past and current threat to honeybee health.” I would have liked to see such “careful consideration” and critical discussion in the paper. Since this vital discussion has not been included, let me do some very basic calculations here.

The maximum activity concentration of ^{137}Cs in honey found in this study was 19 Bq/kg. Honey contains natural potassium, a small fraction of which is radioactive ^{40}K . The natural potassium content of honey ranges between 400 and 35,000 mg/kg (doi: 10.1186/1743-7075-9-61). The resulting activity of ^{40}K in honey ranges from 13 Bq/kg to 1100 Bq/kg. In other words, even the honey with the highest level of ^{137}Cs -contamination, radioactivity-wise, is in the same range like the ^{40}K -activity of the honey with the lowest potassium content. In honey with the highest potassium content, the natural activity of ^{40}K would have exceeded the anthropogenic activity of ^{137}Cs by more than a factor of 50. In other words, if natural potassium does not do harm to bees, an additional ^{137}Cs activity of 1/50 of natural ^{40}K cannot cause any harm either.

It should be noted that ^{40}K has both a much higher beta max. energy and also a much higher gamma-ray energy, so this primitive example probably still follows a rather conservative approach.

Reviewer #2 (Remarks to the Author):

I think the authors have done a good job responding to the reviewer comments

Response to reviewers

Reviewer 1: The paper lacks the necessary discussion of doses to justify claims such as the suggestion that “environmental danger” persists for more than six decades

While there is considerable uncertainty and an active debate on the negative effects of low-dose radiation on insects (Moller & Mousseau, 2013; Hiyama et al., 2013, Taira et al., 2019; Raines et al., 2020, among others cited in our manuscript) we understand this perspective. We have removed “environmental danger” from the abstract and replaced this with the “environmental legacy ...can persist for more than six decades”.

or “the magnification of 137Cs in honey should be carefully considered as a major past and current threat to honeybee health.”

This has been rephrased to, “Given that pollinating insects provide vital services to the world’s ecosystem and are essential in maintaining global food security⁴⁹, more research is needed to help us better understand how ionizing pollution threatens their health and survival” in lines 280-282

I would have liked to see such “careful consideration” and critical discussion in the paper. Since this vital discussion has not been included, let me do some very basic calculations here. The maximum activity concentration of 137Cs in honey found in this study was 19 Bq/kg. Honey contains natural potassium, a small fraction of which is radioactive 40K. The natural potassium content of honey ranges between 400 and 35,000 mg/kg (doi: 10.1186/1743-7075-9-61). The resulting activity of 40K in honey ranges from 13 Bq/kg to 1100 Bq/kg. In other words, even the honey with the highest level of 137Cs-contamination, radioactivity-wise, is in the same range like the 40K-activity of the honey with the lowest potassium content. In honey with the highest potassium content, the natural activity of 40K would have exceeded the anthropogenic activity of 137Cs by more than a factor of 50. In other words, if natural potassium does not do harm to bees, an additional 137Cs activity of 1/50 of natural 40K cannot cause any harm either. It should be noted that 40K has both a much higher beta max. energy and also a much higher gamma-ray energy, so this primitive example probably still follows a rather conservative approach.

We are disappointed that the reviewer didn’t acknowledge that we measured 40K in each of the 122 honey samples alongside 137Cs and provided those data as supplemental information. Instead, they speculated on the 40K activity in our samples and thus underestimated the relative significance of 137Cs. Our samples ranged from 4 to 105 Bq 40K kg⁻¹, somewhat lower than the reviewer assumed, and thus the relative 137Cs contribution to the radiation is meaningful in a portion of our samples. We agree with the reviewer that this comparison is useful and thus provide a short discussion along with a new figure to support our position in the revision. The external dose conversion from Bq/kg to $\mu\text{Gy hr}^{-1}$ shows that in seven cases the ionizing radiation from pollutant 137Cs is equal to or higher than that of 40K (Figure 5), and in a third of the samples the 137Cs sourced >10% of the ionizing radiation. Please see lines 235-283 for a revised discussion including the recommended dose units, and we place our data within a context of the very real debate that is occurring about the potential harmful effects of low-dose radiation on insects